# A Double-Blind, Randomized, Placebo-Controlled Crossover Clinical Study of the Effects of Alpha-s1 Casein Hydrolysate on Sleep Disturbance

**DOI:** 10.3390/nu11071466

**Published:** 2019-06-27

**Authors:** Hyeon Jin Kim, Jiyeon Kim, Seungyeon Lee, Bosil Kim, Eunjin Kwon, Jee Eun Lee, Min Young Chun, Chan Young Lee, Audrey Boulier, Seikwan Oh, Hyang Woon Lee

**Affiliations:** 1Sleep Center, Ewha Womans University Mokdong Hospital, Seoul 07985, Korea; 2Departments of Neurology, Ewha Womans University School of Medicine and Ewha Medical Research Institute, Seoul 07804, Korea; 3Departments of Medical Science, Ewha Womans University School of Medicine and Ewha Medical Research Institute, Seoul 07804, Korea; 4INGREDIA Co., Ltd., CEDEX, 62000 Arras, France; 5Departments of Molecular Medicine, Ewha Womans University School of Medicine and Ewha Medical Research Institute, Seoul 07804, Korea

**Keywords:** alpha-s1 casein hydrolysate, Lactium^®^, nutritional supplements, clinical trial, sleep disturbance, insomnia, sleep quality

## Abstract

This study evaluated the effects of alpha-s1 casein hydrolysate (ACH; Lactium^®^) on the subjective and objective sleep profiles of a community-based sample of Koreans with poor sleep quality. We performed a double-blind, randomized crossover trial with 48 participants (49.0 ± 1.7 years old, 65% female) who exhibited a mild to moderate degree of sleep disturbance. Either ACH or placebo was administered for the initial four weeks, and the counterpart was administered in precisely the same manner after a four-week washout period. Sleep disturbance scales, daytime functioning, and psychiatric aspects showed a similar tendency to improve during both ACH and placebo phases without significant group differences. Overall perceived sleep profiles in sleep diaries were significantly improved during the ACH phase, represented by increased total sleep time and sleep efficiency (SE), as well as decreased sleep latency and wake after sleep onset (WASO). Interestingly, actigraphy demonstrated significantly increased SE after continuous use of ACH for four weeks, clearly more improved when compared to two weeks of use. The polysomnography measures showed a similar tendency without statistically significant group differences. Our findings suggest that refined ACH was well tolerated and could improve sleep quality, with possible cumulative beneficial effects with long-term administration.

## 1. Introduction

Sleep disturbance, a common complaint in the general population, can considerably influence the physical and psychiatric health of individuals and increase social burden due to unwanted errors or accidents. The prevalence of insomnia symptoms has been reported to be up to 48% in Western countries, representing more than one-fifth of the population (22.8%) in Korea; this discrepancy may be due to differences in research criteria and cultural backgrounds [1,2,3].

Currently, cognitive behavioral therapy is the first-line treatment for insomnia disorders; however, hypnotics are widely used and often preferred [4]. Pharmacologic treatment could be a fast and straightforward approach to relieve sleep-related complaints, although it often causes dependence on medication or recurrence of insomnia when the treatment is discontinued [5,6]. Moreover, patients may experience side effects of conventional sleep inducers, which range from morning somnolence to delirium or accidental injuries [7,8]. Therefore, there has been considerable interest in developing alternative treatments, such as dietary pattern intervention or specific nutrient supplements, which may improve the overall quality of life for insomnia patients by lowering the risk of over- or misuse and avoiding the side effects of hypnotics [9,10,11]. Tryptophan and group B vitamins, which are traditional well-known dietary components, are precursors of serotonin, an intermediary product in the production of melatonin [12,13]. Tryptophan has been shown to have direct beneficial effects on the homeostatic regulation of sleep, not only in subjects with insomnia but also those without sleep disturbance have been found to increase sleepiness and shorten sleep latency [14]. Moreover, unlike the photoperiodically regulated production of melatonin in the pineal gland, the release of gastrointestinal melatonin by the enterochromaffin cells appears to be regulated by food intake, particularly after tryptophan loading [15].

Clinical and epidemiological studies have repeatedly demonstrated the comorbid and bidirectional nature of sleep disturbances and psychiatric aspects [16,17,18]. There are common, well-known neurobiological mechanisms (i.e., neurotransmitters and brain structures) associated with both mental disorders and insomnia. One of the most widely accepted mediators in the pathophysiology of anxiety disorders is the gamma-aminobutyric acid (GABA) system, which is also the target of conventional hypnotic drugs, through preferential interaction with α2/α3 and α1 GABA_A_ receptors [19,20]. This suggests that treatments that can alleviate anxiety may provide support for individuals struggling with sleep complaints. Recently, many natural plant-derived substances such as valerian extract (*Valerian* spp.), lemon balm extract (*Melissa officinalis*), passion flower extract (*Passiflora incarnata*), and hops extract (*Humulus lupulus*) have been shown to improve sleep disturbances, with possible mechanisms related to the GABA system [21,22,23,24].

Alpha-s1 casein hydrolysate (ACH), one of the main components of milk protein, has been reported to exhibit antistress effects in addition to blood pressure control, immune control, and antithrombosis effects [25,26,27,28]. In particular, a bioactive decapeptide (α_S1_-casein (f91–100) or α-casozepine), initially believed to exhibit an affinity for GABA_A_ receptors, has been shown to have two flexible tyrosine aromatic rings with similar structures to the classical benzodiazepine aromatic rings, thus demonstrating anxiolytic effects [29,30,31]. Recently, one research group showed that the dose-dependent action of ACH on intracellular chloride ion influx was inhibited by co-administration of bicuculline in vitro, which implies that ACH has an effect on GABA_A_ receptors. Moreover, 150 mg/kg ACH orally administered to mice exhibited a significant difference in pentobarbital-induced sleep-promoting tests and slow wave electroencephalography (EEG) activity, which suggests that it has an effect on the central nervous system [32]. More recently, our group demonstrated that protein expression of the β1 receptor subunit of GABA_A_ in the hypothalamus of rats was increased when 300 mg/kg of ACH was administered, resulting in significantly enhanced total sleep and EEG theta wave during sleep [33]. However, there is a paucity of clinically relevant scientific evidence from human studies of sleep improvement. In previous clinical trials, casein hydrolysate-enriched milk positively impacted sleep and daytime dysfunction quality, and a four-week course of the supplement reduced sleep latency (SL) [34,35,36].

This study was performed to investigate the effects of four weeks of refined ACH on various sleep profiles in a community-based sample of Korean adults with mild to moderate insomnia symptoms. In addition, we evaluated the safety of ACH using subjective and objective methods.

## 2. Materials and Methods 

### 2.1. Participants

Participants experiencing uncomfortable sleep were recruited through the use of online and offline advertisements with the help of a recruiting agency. Sixty participants between 20 and 65 years of age who had definite low subjective sleep quality, as measured by the Pittsburgh Sleep Quality Index (PSQI > 5), were screened [37].

The exclusion criteria included: (1) Severe insomnia based on the Insomnia Severity Index (ISI ≥ 22) [38]; (2) history of a disorder affecting sleep quality, such as narcolepsy, obstructive sleep apnea (OSA), restless leg syndrome (RLS), periodic limb movement syndrome (PLMS), or a psychiatric disorder, including depression; (3) events that could cause severe stress within 2 weeks of the first visit (e.g., death of spouse, family issues, legal issues, financial crisis, or immigration); (4) history of use of medication that could influence sleep patterns, including health products or oriental herbs, within 1 month of the first visit; (5) current use of hormone therapy; (6) binge drinking (>140 g/week, 2.5 bottles/week of alcohol, 2.5 shots/day); (7) heavy smoking (>10 cigarettes/day); (8) high caffeine intake (>10 glasses/day); (9) work schedule that causes irregular sleep patterns (e.g., night shift); (10) history of travel to a different time zone within 1 month of the first visit; (11) extremely low or high body mass index (BMI ≤ 18 kg/m^2^ or ≥35 kg/m^2^); (12) history of allergic reaction to milk, milk-containing food, or any of the components in the test product and placebo; (13) history of clinical trial participation within 1 month of the first visit; (14) currently pregnant or breastfeeding; and (15) individuals who were deemed incompatible with the test protocol.

One participant was excluded due to use of sleeping pills, and 11 participants withdrew their consent before randomization. Therefore, 48 participants were included in the randomization step (Figure 1). This study was approved by the Institutional Review Board (IRB) of Ewha Womans University Mokdong Hospital and was registered with the Clinical Research Information Service (CRIS; study number KCT0001867).

### 2.2. Study Design

We performed a double-blind, randomized, placebo-controlled crossover trial. The duration of this study for the 48 participants was 12 weeks. Either a test or placebo capsule was administered during the initial 4 weeks of phase I; after a 4-week washout period, the counterpart capsule was administered during the 4 weeks of phase II in the exact same manner.

Twenty-four participants were randomly assigned to each test and control group for phase I. While we screened for allergy history, one participant in the control group was suspended due to an adverse event. Two participants in the test group withdrew their consent after the first randomization. Two participants completed phase I in the control group but withdrew their consent during the washout period. Finally, 43 participants completed both phase I and phase II of the study.

For the safety evaluation, vital signs were taken and routine blood tests were performed. Additionally, individuals were allowed to maintain their habits during the entire study period, but dairy products and high-tryptophan foods were restricted. To this end, we provided a list of foods to avoid (Appendix A), and we checked compliance through a survey of diet diaries.

### 2.3. Tested Products

Bovine ACH (Lactium^®^, Ingredia, Arras, France) was the test product used in the test phase, while the placebo product used in the control phase was resistant maltodextrin (Novarex Co., Ltd., Cheongju, Korea). Based on the safety analysis in animal studies, the safe intake amount was calculated to be 300 mg/day of Lactium^®^ for this study. The test capsule was composed of 75% ACH (300 mg), 24% maltodextrin, and 1% silicon dioxide. ACH was composed of 2.2% (6.6 mg) α_S1_-casein (f91-100), which was detected by using high-performance liquid chromatography and photometric diode array. The placebo capsule was composed of 99% maltodextrin and 1% silicon dioxide. The capsules were indistinguishable from each other by either weight (400 mg) or morphology. Participants were asked to take either the test or placebo capsule every day an hour before their bedtime.

### 2.4. Sleep Quantity and Quality Assessment

#### 2.4.1. Sleep and Mood Questionnaire Scales

We continuously monitored the participants’ subjective sleep quality and psychiatric characteristics through a survey consisting of various questionnaire sets at 2-week intervals. We used PSQI and ISI in the monitoring phase and the screening phase to conduct sleep assessment, including insomnia symptoms. Subjective excessive daytime sleepiness (EDS) was measured using the Epworth Sleepiness Scale (ESS), a widely used 8-item questionnaire regarding participants’ likelihood of falling asleep in different situations [39]. We also assessed participants’ fatigue symptoms through the Fatigue Severity Scale (FSS) [40]. Depression and anxiety symptoms were measured with the Beck Depression Inventory (BDI) and Beck Anxiety Inventory (BAI), respectively; higher scores reflected higher levels of symptoms [41,42].

#### 2.4.2. Sleep Diary

Participants were asked to record 8 weeks of a sleep diary regarding the times when they went to sleep and woke up, in order to determine time in bed (TIB), time taken to fall asleep (SL), subjective total sleep time (TST), number and time of each waking after sleep onset (WASO), reasons for such disruptions, and their feelings in the morning. Daily sleep diary data were averaged for one week, and sleep efficiency (SE) was calculated as the percentage of TST compared to total TIB.

#### 2.4.3. Actigraphy

Participants used actigraphy on the nondominant hand during the 8 weeks of recording for both phase I and phase II (Actiwatch-2 or Actiwatch Spectrum PRO, Philips Respironics, Murrysville, PA, USA). The variables measured were TIB, SL, TST, SE, and WASO. The standard for normal SE was set at ≥85%.

#### 2.4.4. Polysomnography (PSG)

Full-night PSG was performed before and after each phase on 24 subjects who agreed to the one-night stay at the sleep laboratory to take the sleep test in the hospital. The PSG used the Grass Telefactor (Beehive Horizon, Montreal, QC, Canada) device, including 6 EEG channels (F3-A2, F4-A1, C3-A2, C4-A1, O1-A2, O2-A1), 1 jaw electromyograph (EMG) channel, 4 electrooculogram (EOG) channels (LE-A2, RE-A1, SO-A2, IO-A2), 2 breathing sensor channels (chest and abdomen), 1 oxygen saturation monitor channel, electrocardiogram (ECG), 2 anterior tibialis EMG channels, and a position monitoring channel. PSG data were analyzed according to the standard guidelines of the American Academy of Sleep Medicine (AASM) [43].

### 2.5. Statistical Methods

The sample size was estimated to be 48 subjects, yielding 80% power to detect an effect (Cohen’s d = 0.47) with a two-tailed *t-*test at a 0.05 significance level. The estimated change of the objective TST was based on previously published data [44]. An intention-to-treat (ITT) analysis was conducted involving all randomized participants. A normality test was performed to analyze the nonnormally distributed data after being converted to a normal distribution. At baseline, participant characteristics were described as mean and standard deviation (SD) for continuous variables and frequency for categorical variables. Student’s *t*-test was applied to compare the compliance and safety (vital signs, routine blood tests) between ACH and placebo groups. To evaluate the sleep and psychiatric profiles, a linear mixed-effects model was applied with group, time, group × time, sequence, and period as the fixed effects and participant as the random effect. Statistical analysis was performed using SAS version 9.4 (SAS Institute Inc., Cary, NC, USA). A two-tailed value of *p* < 0.05 was considered to be significant. Corrected *p*-value (or *q*-value) was also calculated using Storey’s false discovery rate (FDR) approach (95% confidence interval) to correct for multiple comparisons.

## 3. Results

### 3.1. Demographic Characteristics

The demographic characteristics of the 48 participants who were enrolled at the time of randomization are presented in Table 1. The average age was 49.0 ± 11.9 years old. There were 17 men and 31 women. The mean sleep disturbance duration of participants was 57.6 ± 13.2 months, and their baseline PSQI and ISI scores were 11.4 ± 1.9 and 13.2 ± 3.8, respectively. After exclusions and losses, a total of 43 participants were eligible for analysis. The compliance of ACH and placebo groups was above 90%, and there was no significant difference between the two groups (94.0 ± 6.6 and 90.0 ± 16.4, respectively, *p* = 0.136).

### 3.2. Sleep and Mood Questionnaire Scales

Differences in sleep disturbance symptoms, daytime functioning, and psychiatric characteristics are presented in Table 2. The ACH phase showed a gradual decrease in PSQI score, which was 8.79 ± 0.43 after two weeks and 8.51 ± 0.43 after four weeks, compared with the baseline score of 9.79 ± 0.42. During the placebo phase, PSQI also decreased from baseline to the end of the phase, thus change in both groups was in the direction of improved sleep (*p* = 0.211). The sleep discomfort index (measured by ISI) was similar, with a significant effect of time (*p* < 0.001), although there was no statistically significant overall difference compared to the placebo phase (*p* = 0.523).

Participants’ daytime functioning also gradually improved during the four-week administration period; however, there were no significant differences between phases (ESS, *p* = 0.324; FSS, *p* = 0.854). Regarding depression and anxiety indicators, participants exhibited a similar tendency to improve during both ACH and placebo phases without significant group differences (BDI, *p* = 0.912; BAI, *p* = 0.924).

### 3.3. Subjective and Objective Sleep Profile Monitoring

Sleep profile monitoring during the study was performed using both subjective (sleep diary) and objective (actigraphy) methods; these results are presented in Table 3 and Table 4. TIB was decreased in both placebo and ACH phases, slightly more during the placebo period, without statistical significance (sleep diary, *p* = 0.465; actigraphy *p* = 0.656). TST increased and SL decreased during the ACH test phase, indicating a tendency toward significantly improved sleep quantity; in contrast, a worsening tendency was observed during the placebo control phase by both measures. Statistically significant differences were observed in the sleep diary assessment, but not in the actigraphy assessment (TST, *p* < 0.001, *q* < 0.001 vs. *p* = 0.270; SL, *p* < 0.001, *q* < 0.001 vs. *p* = 0.063). The ratio of participants with a tendency toward improved sleep quantity was also higher in the ACH phase compared to the placebo phase by both sleep diary and actigraphy measures.

SE was significantly improved during the test phase compared to the control phase by both measures (Figure 2). The effect was not apparent during the first two weeks (*p* = 0.067 by sleep diary and *p* = 0.066 by actigraphy), but a definite improvement in SE was noticeable after continuous use of ACH for four weeks (*p* < 0.001, *q* < 0.001 by sleep diary and *p* = 0.007, *q* = 0.016 by actigraphy). The ratio of participants with increased SE was also higher in the ACH phase compared to the placebo phase (85% vs. 24%, *p* < 0.001, *q* < 0.001 by sleep diary; 85% vs. 63%, *p* = 0.031, *q* = 0.078 by actigraphy).

Sleep disturbance measures determined by WASO and number of awakenings showed a declining tendency during the test phase; this also indicated sleep quality improvement (WASO, *p* < 0.001, *q* < 0.001 by sleep diary vs. *p* = 0.053 by actigraphy; number of awakenings, *p* = 0.077 by sleep diary vs. *p* = 0.240 by actigraphy). The ratio of participants with disturbed sleep was lower in the test phase than in the placebo phase by both measures.

### 3.4. PSG Measures

The 24 participants who completed overnight PSG measurements before and after each phase were postmenopausal women. The majority of PSG measures showed similar changes in the ACH and placebo groups from baseline to the end of treatment (Appendix A). TST and SE were increased in both phases, slightly more during the ACH period, without statistical significance (TST, *p* = 0.729; SE, *p* = 0.599). SL and WASO decreased in both phases, slightly more during the ACH period; however, there was no particular statistical significance (SL, *p* = 0.437; WASO (min), *p* = 0.551). The number of awakenings and arousal index exhibited no significant difference between the phases.

The overall sleep structure indicated that the overall percentage of non-rapid eye movement (NREM) sleep decreased and rapid eye movement (REM) sleep increased after both test and control phases, without significant differences (NREM, *p* = 0.483; REM, *p* = 0.440). Specifically, the percentage of stage 2 sleep was relatively increased after the ACH phase, whereas the percentage of slow wave sleep (SWS) was increased after the placebo phase (stage 2, *p* = 0.116; SWS, *p* = 0.156).

There were also no significant group differences between baseline and endpoint on polysomnography measures such as the apnea-hypopnea index (AHI) and PLM index (*p* = 0.661 and *p* = 0.749, respectively).

### 3.5. Safety Evaluation

There was one adverse event (itching and urticaria) reported during the initial phase in the placebo group. No participants assigned to the ACH group during phases I and II reported adverse events. In order to evaluate the safety of ACH, we assessed hematological profiles in addition to vital signs before and after each control and test phase (Appendix A). ACH administration did not affect any vital signs or laboratory examinations.

## 4. Discussion

Sleep is one of the most basic and vital biological processes and is essential for maintaining physical health and mental stability. Sleep deprivation impairs daytime performance and diminishes quality of life, and insomnia now represents a major public health concern worldwide by causing metabolic, cardiovascular, and neurodegenerative diseases [13]. This study was performed to investigate the effects of a dairy product with a specifically refined ACH ingredient on various sleep profiles in Korean adults with poor sleep quality, based on the known anxiolytic-like effect of the product [26,28]. It is known that stress and anxiety, in particular, cause changes in the sleep/wake rhythm and interfere with both sleep quantity and quality, including SE and architecture [45,46,47].

Sleep quality is a complex construct to evaluate empirically, and dissatisfaction with sleep quantity or quality could be a result of various factors, including difficulty initiating or maintaining sleep, early morning awakening, or nonrestorative sleep [48,49]. Previous studies have shown that the PSQI is a valuable tool for assessing perceived sleep quality; however, its measurements can be quite different from objective reality [50]. PSG is the gold standard for objective measurement of sleep stages and architecture; however, it is a limited method in the hospital or laboratory setting and may not accurately reflect the ordinary continuous sleep patterns of individuals who are sensitive to unfamiliar surroundings or who already have insomnia symptoms. Therefore, we monitored sleep behavior patterns and related clinical symptoms using actigraphy as well as questionnaire scales and sleep diaries in order to comprehensively investigate whether ACH supplementation could help individuals with sleep disturbances.

The strengths of this randomized placebo-controlled trial are its double-blind crossover design and the thorough sleep investigation by subjective and objective methods; moreover, it was conducted with a refined ACH capsule with a relatively high dosage (300 mg/day). Some studies have verified the efficacy of ACH without a crossover design, only with subjective symptom scales [34,35]. One clinical ACH trial with both sleep diaries and actigraphy measurements showed beneficial effects concerning TST and efficiency; however, their tested component was casein hydrolysate-enriched milk combined with other ingredients [36]. This is important not only to determine whether the specific ACH ingredient is beneficial for sleep, but it can also be an important result for many adults with lactose intolerance who cannot drink milk [51].

Our data, measured by subjective symptom scales including PSQI, ISI, ESS, FSS, BDI, and BAI, also demonstrated gradual improvement during the ACH phase but did not show a statistically significant difference compared to the placebo phase. Potential effects of ACH on applied subjective symptom scales might be too marginal to exceed the placebo response in the present double-blind trial with 48 participants. In addition, it is possible that the behavior of keeping a sleep diary itself may affect the subjective sleep and psychiatric parameters because it is the first stage in the clinical intervention of chronic insomnia [52]. Moreover, since the average period of sleep disturbance of our participants was over six years (72.0 ± 13.2 months), four-week administration of ACH may not have been enough to produce a significant result. However, we showed the statistically significant efficacy of ACH administration by improved perceived sleep profiles in the sleep diary records and confirmed increased SE objectively by actigraphy measurement. Besides, PSG measurement in a subgroup showed that there were no effects on the parameters of primary sleep disorders, such as sleep apnea or periodic leg movements.

Interestingly, the effects of ACH on SE were more obviously demonstrated after four weeks of consumption compared to two weeks, which suggests that ACH can improve sleep quality when administered for an extended period without substantial side effects or tolerance issues. This finding can be a significant advantage of dietary supplements relative to other conventional hypnotic medications, which often show reduced effects or tolerance after long-term use [53]. Further prospective follow-up studies with more extended administration periods should be considered for more specific assessment of cumulative ACH effects without tolerance or dependence issues.

## 5. Conclusions

We performed a randomized, placebo-controlled trial with a double-blind crossover design in a community-based sample of adults with low sleep quality, using a daily supplement of highly standardized ACH (Lactium^®^) for sleep profile improvement. The regimen of 300 mg/day of ACH for four weeks was able to significantly improve subjective sleep quantity and quality, based on increased TST and SE, as well as reduce SL and WASO. Objective sleep parameters by actigraphy also showed significantly increased SE, which was more clearly demonstrated after continuous use for four weeks compared to two weeks. These results suggest that ACH can be safely used to help individuals with sleep disturbances, especially for mild to moderate insomnia symptoms.

## Figures and Tables

**Figure 1 nutrients-11-01466-f001:**
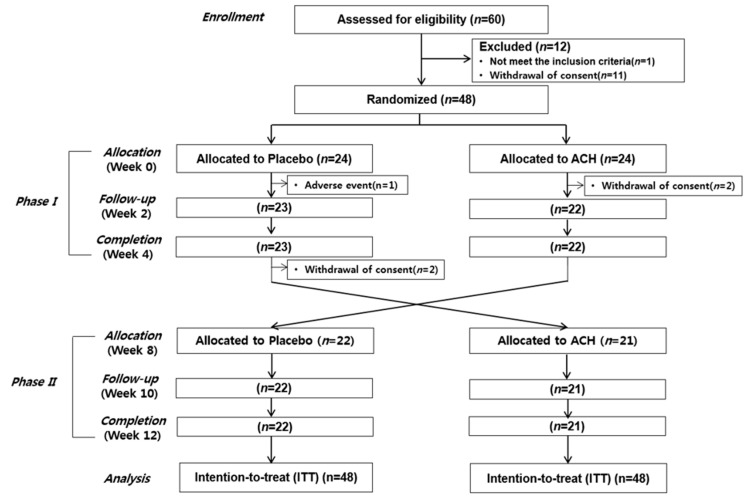
Study protocol. ACH, alpha-s1 casein hydrolysate.

**Figure 2 nutrients-11-01466-f002:**
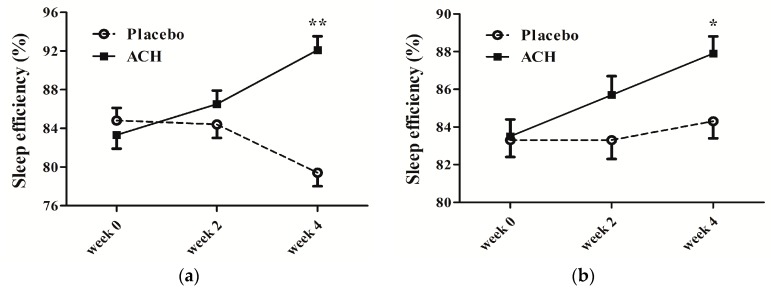
Comparison of effects of ACH administration duration on sleep efficiency. (**a**) Sleep diary showed constant improvement in sleep efficiency for ACH phase group compared with gradual deterioration for placebo group. Group difference in sleep efficiency over time became significant when ACH was administered for four weeks (*p* < 0.001, *q* < 0.001 for 0–4 weeks), which did not reach statistical significance after two weeks of use (*p* = 0.067 for 0–2 weeks). (**b**) Actigraphy revealed that sleep efficiency did not show clear differences between ACH and placebo phases; however, the significant improvement effect was only obvious after continuous use of ACH for four weeks (*p* = 0.066 for 0–2 weeks vs. *p* = 0.007, *q* = 0.016 for 0–4 weeks). * *q*-value < 0.05 and ** *q*-value < 0.01, pFDR was calculated for group × time difference by linear mixed-effects model to account for multiple testing.

**Table 1 nutrients-11-01466-t001:** Demographic characteristics of participants.

Variables	N = 48
Age (year)	49.0 ± 11.9
Gender (male/female)	17/31
Menopause	24/31 (77.4%)
Postmenopausal period (month)	72.0 ± 57.2
Sleep disturbance duration (month)	57.6 ± 91.3
Pittsburgh Sleep Quality Index	11.4 ± 1.9
Insomnia Severity Index	13.2 ± 3.8
Family history of insomnia	6 (12.5%)
Caffeine amount (servings/day)	1.2 ± 1.3
Alcohol drinker	31 (64.6%)
Alcohol amount (g/week)	29.9 ± 37.4
Smoker	2 (4.2%)
Smoking amount (cigarettes/day)	0.3 ± 1.1
Anthropometric measures	
Body weight (kg)	60.3 ± 10.9
Body mass index (kg/m^2^)	22.8 ± 2.8
Waist circumference (cm)	80.0 ± 7.9
Neck circumference (cm)	33.9 ± 2.8
Vital signs	
Systolic blood pressure (mmHg)	119.4 ± 12.8
Diastolic blood pressure (mmHg)	74.5 ± 12.1
Pulse rate (beats/min)	71.2 ± 12.3
Body temperature (°C)	36.4 ± 0.3

Data are mean ± SD (standard deviation) or number of answers (%).

**Table 2 nutrients-11-01466-t002:** Differences in sleep disturbance symptoms, daytime functioning, and psychiatric aspects between control (placebo) and test (alpha-s1 casein hydrolysate (ACH)) phases.

Variables	Placebo			ACH			*p*-Value ^†^		
							Group	Time	Group × Time
Pittsburgh Sleep Quality Index									
Week 0	10.14	±	0.42	9.79	±	0.42			
Week 2	8.23	±	0.42	8.79	±	0.43			
Week 4	8.41	±	0.42	8.51	±	0.43	0.668	<0.001 **	0.211
Insomnia Severity Index									
Week 0	12.15	±	0.73	12.08	±	0.73			
Week 2	9.75	±	0.73	10.50	±	0.74			
Week 4	9.44	±	0.73	10.04	±	0.74	0.406	<0.001 **	0.523
Epworth Sleepiness Scale									
Week 0	6.30	±	0.58	6.72	±	0.58			
Week 2	6.10	±	0.58	5.91	±	0.59			
Week 4	6.02	±	0.58	5.67	±	0.59	0.920	0.039	0.324
Fatigue Severity Scale									
Week 0	31.60	±	1.64	32.55	±	1.65			
Week 2	31.00	±	1.65	32.64	±	1.67			
Week 4	30.67	±	1.65	31.29	±	1.67	0.276	0.462	0.854
Beck Depression Inventory									
Week 0	11.65	±	1.25	12.37	±	1.26			
Week 2	11.49	±	1.26	12.28	±	1.27			
Week 4	10.41	±	1.26	10.79	±	1.27	0.464	0.008 *	0.912
Beck Anxiety Inventory									
Week 0	8.04	±	1.11	7.71	±	1.11			
Week 2	8.41	±	1.11	7.72	±	1.12			
Week 4	7.70	±	1.11	7.09	±	1.12	0.363	0.362	0.924

Data are least square (LS) mean ± standard error of mean (SEM). **^†^**
*p*-value, linear mixed-effects model was used to analyze the effects of group, time, and group × time for four weeks. * *q*-value < 0.05 and ** *q*-value < 0.01, Storey’s positive false discovery rate (pFDR) was calculated to account for multiple testing.

**Table 3 nutrients-11-01466-t003:** Comparison of sleep parameters between control (placebo) and test (ACH) groups.

Variables	Placebo		ACH		*p*-Value ^†^		
	Week 0	Week 4	Week 0	Week 4	Group	Time	Group × Time
TIB (min)							
Sleep diary	470.4 ± 12.1	455.9 ± 12.4	464.5 ± 12.2	459.5 ± 12.2	0.883	0.134	0.465
Actigraphy	432.6 ± 17.0	412.0 ± 17.1	431.5 ± 17.1	424.0 ± 17.5	0.737	0.342	0.656
TST (min)							
Sleep diary	395.7 ± 10.0	361.0 ± 10.3	385.3 ± 10.1	422.7 ± 10.1	<0.001 **	0.796	<0.001 **
Actigraphy	362.8 ± 16.5	347.8 ± 16.6	360.1 ± 16.6	376.1 ± 17.0	0.414	0.971	0.270
SL (min)							
Sleep diary	33.2 ± 4.4	50.5 ± 4.6	39.5 ± 4.5	18.3 ± 4.5	0.011 *	0.598	<0.001 **
Actigraphy	4.4 ± 1.5	5.5 ± 1.5	7.0 ± 1.5	2.8 ± 1.6	0.975	0.288	0.063
SE (%)							
Sleep diary	84.9 ± 1.3	79.4 ± 1.3	83.3 ± 1.3	92.1 ± 1.3	<0.001 **	0.108	<0.001 **
Actigraphy	83.3 ± 0.9	84.3 ± 1.0	83.6 ± 0.9	88.0 ± 1.0	0.013 *	<0.001 **	0.007 *
WASO (min)							
Sleep diary	15.7 ± 3.4	29.6 ± 3.6	17.0 ± 3.5	11.9 ± 3.5	0.039	0.097	<0.001 **
Actigraphy	53.2 ± 3.8	49.2 ± 3.9	55.6 ± 3.9	38.9 ± 3.9	0.228	0.002 *	0.053
Awake (N)							
Sleep diary	1.0 ± 0.1	1.1 ± 0.1	0.9 ± 0.1	0.8 ± 0.1	0.001 **	0.216	0.077
Actigraphy	16.1 ± 1.1	15.2 ± 1.1	15.6 ± 1.1	12.6 ± 1.2	0.157	0.024	0.240

Data are LS mean ± SEM. **^†^**
*p*-value, linear mixed-effects model was used to analyze the effects of group, time, and group × time for four weeks. * *q*-value < 0.05 and ** *q*-value < 0.01, pFDR was calculated as to account for multiple testing. TIB, time in bed; TST, total sleep time; SL, sleep latency; SE, sleep efficiency; WASO, wake after sleep onset.

**Table 4 nutrients-11-01466-t004:** Ratio of participants with improved sleep parameters in control (placebo) and test (ACH) phases.

Variables	Sleep Diary			Actigraphy		
	Placebo	ACH	*p*-Value ^†^	Placebo	ACH	*p*-Value ^†^
Increased TST	24%	79%	<0.001 **	41%	61%	0.073
Decreased SL	30%	67%	0.001 **	29%	46%	0.119
Increased SE	24%	85%	<0.001 **	63%	85%	0.031
Decreased WASO	32%	51%	0.096	61%	67%	0.597
Decreased Awakes	43%	54%	0.355	56%	64%	0.465

Data are answered numbers (%). **^†^**
*p*-value, chi-square test was used to analyze group difference. * *q*-value < 0.05 and ** *q*-value < 0.01, pFDR was calculated to account for multiple testing.

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
