# Peer review of "A Double-Blind, Randomized, Placebo-Controlled Crossover Clinical Study of the Effects of Alpha-s1 Casein Hydrolysate on Sleep Disturbance"

_nutrients, 2019, doi:10.3390/nu11071466_

Reviewer 1 Report

This study aimed to investigate the effect of alpha-s1 casein hydrolysate, a dietary supplement derived from dairy product, on sleep among middle aged participants with sleep disturbances. This is a well-designed double-blind randomized crossover trial with sophisticated measurements of sleep. Randomized studies regarding the effect of nutrient supplements on sleep are scarce. Therefore, the study is important concerning its novelty and potential utility in the development of treatment alternatives for insomnia symptoms and sleeplessness. However, certain limitations apply to the present manuscript, and the authors are requested to take the comments below into consideration when making a revision.

1. Introduction: Some extra points are worth mentioning here: 1) additional evidence of nutrient supplements improving sleep, see St-Onge et al., Effects of Diet on Sleep Quality. Adv Nutr. 2016. 2) Effect of dietary intervention on sleep among insomniacs, see Tan et al., Effect of Six-Month Diet Intervention on Sleep among Overweight and Obese Men with Chronic Insomnia Symptoms: A Randomized Controlled Trial, Nutrients. 2016. 3) Possible mechanisms underlying dairy product, tryptophan, and sleep quality, see Peuhkuri et al., Diet promotes sleep duration and quality. Nutr Res. 2012.

2. Ref 27, please give detailed information about the dose impact of ACH on counterbalancing the effect of GABA receptor antagonist. Was the ACH administration in animal models targeting on central or peripheral neurons?

3. The consumption of dairy products of the participants should be considered in the analysis. If available, authors are encouraged to control this variable when making the comparisons.

4. What was the timing and frequency of ACH/placebo administration in this trial (e.g. before sleep, once a day)?

5. In the placebo group, there was a clear trend of sleep quality deterioration as determined by sleep diary and actigraphy. However, this scenario was totally opposite when looking at the results in PSQI, ISI and other subjective scales. What might be the cause of this drastic discrepancy? Ref 45 reported the inconsistency between PSQI and actigraphic sleep data among older adults, however it was unavailable to suggest whether the trend between these measurements show the same discrepancy as demonstrated in the present study. More studies should be involved in this section of discussion.

Author Response

1. Introduction: Some extra points are worth mentioning here

1) Additional evidence of nutrient supplements improving sleep, see St-Onge et al., Effects of Diet on Sleep Quality. Adv Nutr. 2016.

2) Effect of dietary intervention on sleep among insomniacs, see Tan et al., Effect of Six-Month Diet Intervention on Sleep among Overweight and Obese Men with Chronic Insomnia Symptoms: A Randomized Controlled Trial, Nutrients. 2016.

3) Possible mechanisms underlying dairy product, tryptophan, and sleep quality, see Peuhkuri et al., Diet promotes sleep duration and quality. Nutr Res. 2012.

Author response >

We reviewed the recommended papers and added them to the revised manuscript as follows.

Page 2, Line 50: Therefore, there has been considerable interest in the development of alternative treatments, such as dietary pattern intervention or specific nutrient supplements, which may improve the overall quality of life for insomnia patients by lowering the risks of over/misusage and avoiding the side effects of hypnotics [9-11]. Tryptophan and group B vitamins are the traditionally well-known dietary components, which are the precursor of serotonin, an intermediary product in the production of melatonin [12, 13].

Reference>

11.         Tan, X.; Alén, M.; Wang, K.; Tenhunen, J.; Wiklund, P.; Partinen, M.; Cheng, S. Effect of six-month diet intervention on sleep among overweight and obese men with chronic insomnia symptoms: a randomized controlled trial. Nutrients 2016, 8, 751.

12.        Peuhkuri, K.; Sihvola, N.; Korpela, R. Diet promotes sleep duration and quality. Nutrition research 2012, 32, 309-319.

13.        Kyle, S.D.; Morgan, K.; Espie, C.A. Insomnia and health-related quality of life. Sleep medicine reviews 2010, 14, 69-82.

2. Ref 27, please give detailed information about the dose impact of ACH on counterbalancing the effect of GABA receptor antagonist. Was the ACH administration in animal models targeting on central or peripheral neurons?

Author response >

We have modified the description as below by referring to the comments.

Page 2, Line 72: Recently, one research group showed the dose-dependent ACH action on the intracellular chloride ion influx was inhibited by co-administration of bicuculline in vitro, which implies the effect of ACH on the GABAA receptor. Moreover, the orally administered 150mg/kg ACH to mice exhibited a significant difference in pentobarbital-induced sleep promoting test and slow wave electroencephalography (EEG) activity, which suggests its effects on the central nervous system [30]. More recently, our group demonstrated that protein expression of the β1 receptor subunit of GABAA in the hypothalamus of rats was increased when ACH of 300mg/kg was administered, resulting in significantly enhanced total sleep and EEG theta wave during sleep [31].

Reference >

30.        dela Pena, I.J.I.; Kim, H.J.; de la Pena, J.B.; Kim, M.; Botanas, C.J.; You, K.Y.; Woo, T.; Lee, Y.S.; Jung, J.-C.; Kim, K.-M. A tryptic hydrolysate from bovine milk αs1-casein enhances pentobarbital-induced sleep in mice via the GABAA receptor. Behavioural brain research 2016, 313, 184-190.

31.        Yayeh, T.; Leem, Y.H.; Kim, K.M.; Jung, J.C.; Schwarz, J.; Oh, K.W.; Oh, S. Administration of Alphas1-Casein Hydrolysate Increases Sleep and Modulates GABAA Receptor Subunit Expression. Biomol Ther (Seoul) 2018, 26, 268-273, doi:10.4062/biomolther.2017.083.

3. The consumption of dairy products of the participants should be considered in the analysis. If available, authors are encouraged to control this variable when making the comparisons.

Author response >

We instructed all of the participants to limit the consumption of tryptophan high-foods such as dairy products, soybeans and related products, egg and poultry. We have added a detailed description of the restricted type of food as an additional supplementary materials (table S1) in the revised manuscript as follows.

Page 4, Line 128: Additionally, individuals were allowed to maintain their habits during the entire study period, but dairy products and high-tryptophan concentration foods were restricted. To this end, we provided an avoiding list (table S1), and we checked compliance through a diet diary survey.

Page 11, Line 346: Table S1: The list of dairy products and high-tryptophan concentration foods that were restricted during the study.

Table S1. The list of dairy products and high-tryptophan concentration foods that were restricted during the study.

Category

List   of foods

Do   not eat:

Dairy products

Milk, Yogurt, Cheese

Pork   Tripe/Chitterlings

Grilled tripe, Tripe hotpot

Beans   and related products

Soybean, Tofu, Soy milk, Soy   protein powder

Others

Chi-seed, Spirulina

Limited   (100g/day):

Eggs

2 eggs, 10 quail eggs

Poultry

1 chicken breast, 1 chicken   leg, 10 smoked duck slices  

Meat

Half portion of pork, 2 piece   of tenderloin of lamb/veal

Processed   meat

Half of spam, 10 small Vienna   Sausages, 10 strips of bacon

Nuts

1 cup of peanut, pumpkin seed, sunflower   seed, sesame

Fish

10 slices of smoked salmon, 1   piece of cod, 3/4 can of sea snail

4. What was the timing and frequency of ACH/placebo administration in this trial (e.g. before sleep, once a day)?

Author response >

The method of taking the tested capsule is described in detail as follows

Page 4, Line 139: Participants were asked to take either test or placebo capsule once a day an hour before their bedtime.

5. In the placebo group, there was a clear trend of sleep quality deterioration as determined by sleep diary and actigraphy. However, this scenario was totally opposite when looking at the results in PSQI, ISI and other subjective scales. What might be the cause of this drastic discrepancy? Ref 45 reported the inconsistency between PSQI and actigraphic sleep data among older adults, however it was unavailable to suggest whether the trend between these measurements show the same discrepancy as demonstrated in the present study. More studies should be involved in this section of discussion.

Author response >

Thank you to raise this important issue. We modified and supplemented the discussion section according to the advice.

Page 10, Line 315: Our data measured by subjective symptom scales - PSQI, ISI, daytime functioning and mood questionnaires - also exhibited gradual improvement during the ACH phase but did not show a statistically significant difference compared to the placebo phase. Subjective symptom scales in a double-blind trial could be insufficient to exhibit a significant difference between the placebo and tested ingredient due to placebo response. Also, it is possible that the behavior of keeping a sleep diary itself may have affected the subjective sleep and psychiatric parameters because it is the first stage of the clinical intervention of chronic insomnia [50].

Reference >

50.        Lund, H.G.; Rybarczyk, B.D.; Perrin, P.B.; Leszczyszyn, D.; Stepanski, E. The discrepancy between subjective and objective measures of sleep in older adults receiving CBT for comorbid insomnia. Journal of clinical psychology 2013, 69, 1108-1120.

Reviewer 2 Report

The authors report on an interesting study that addresses the effects of alpha-s1 casein hydrolysate (ACH) on various objective and subjective sleep measures. They carried out a randomized placebo-controlled cross-over trial with 48 subjects experiencing mild to moderate sleep disturbances. Outcome variables included time in bed, total sleep time, sleep latency, sleep efficiency, and WASO as assessed using sleep diaries and actigraphy devices. The authors also made use of polysomnography. Further outcome measures included the Pittsburgh Sleep Quality Index, the Insomnia Severity Index, the Epworth Sleepiness Scale, and the Fatigue Severity Scale. Intention-to-treat analyses were performed by applying linear mixed models. Analysis did not reveal effects of ACH on sleep disturbance symptoms, daytime functioning, psychiatric aspects, and polysomnography parameters. However, analyses revealed convincing effects on sleep diary variables during the intake of ACH relative to the placebo-phase. Actigraphy variables showed similar result patterns, although significant evidence was only revealed for variable ‘sleep efficiency’. It should be mentioned that the latter finding does not appear to withstand a correction for multiple testing (at least not a Bonferroni-correction).

Major strengths of the study are the placebo-controlled crossover design and the use of both objective and subjective sleep measures. A further strength is that it has been registered with the Korean Clinical Research Information Service (study number KCT0001867) that is a primary registry of the WHO International Clinical Trials Registry Platform. A weakness is that the authors do not report on expected or clinically relevant effect sizes, respectively, and how they determined their sample size. Especially in terms of the objective sleep variables (which are interpreted to be consistent with the subjective outcomes though non-significant in the vast majority of tested associations), a larger sample size could have enhanced result reliability. A further weakness is that no correction for multiple testing was applied.

Overall, this manuscript is definitely among the better articles that I usually read as a reviewer. The writing is concise and straightforward and the topic is surely of interest for the readers of this journal. The sleep diary results are convincing and support the authors conclusions. I also feel that the presented study may stimulate further research in this field. Before disseminating this manuscript to the scientific community, a number of issues require further consideration, which I will outline below:

- As mentioned above, the authors should provide more information on how they came up with the selected sample size. A power calculation / sensitivity analysis would be appreciated.

- I generally encourage the authors to apply multiple testing corrections, e.g. by calculating the corresponding FDR value for each primary outcome test. This can easily be done in R based on the derived p values.

- Table 1 shows demographic characteristics of the study sample. For descriptive statistics I recommend to provide standard deviations or ranges instead of standard errors.

- The authors should provide effect size estimates (e.g. eta squared), at least for the significant primary outcome results.

- Figure 2 appears to show a significant impairment of subjective sleep efficiency in the placebo phase. However, the authors state that ‘placebo phase failed to show any effects’. The authors should reconsider the effect of time in the placebo phase and may provide an interpretation why sleep efficiency possibly got worse (‘by chance’ could be one explanation).

- Table 3, minor issue: A more appropriate caption might be ‘comparison of sleep parameters […]’ instead of ‘differences in sleep parameters […]’

- Table 4: The authors may define ‘improved sleep’ more clearly. Was there a certain degree of improvement necessary or did the direction of change – irrespective of magnitude – constitute the decisive criterion?

- Although the manuscript is generally well written, I feel that the grammar and style of the discussion section could be further improved. Some examples for odd sentences:

‘This study is the first clinical, randomized, placebo-controlled trial with both double-blind and crossover design’ – the first addressing sleep effects of ACH?

‘There were some simple double-blind studies revealed the efficacy ACH ingredient by PSQI improvements […].’

‘Because it was a method of measuring subjective symptoms in a double-blind trial, the difference between the two groups may not have been significant due to placebo response.’

 ‘There was one clinical ACH trial with both sleep diary, and actigraphy measurements showed beneficial effects concerning TST and efficiency; […]’

Regarding the discussion section, the authors may consider another English native speaker for proof-reading. In sum, I think this is a very well-designed study that should be disseminated to the scientific community after addressing the issues mentioned above.

Author Response

 1. As mentioned above, the authors should provide more information on how they came up with the selected sample size. A power calculation / sensitivity analysis would be appreciated.

Author response >

The sample size was calculated by referring to the previously published Human RCT comparing the change of total sleep time (TST) by functional food ingredients. TST decreased by 12 minutes in the control group, compared to the 34 minutes increase in the test group when measured by actigraphy. The sample size was calculated to be 18 per sequence, assuming the average standard deviation as 97. In consideration of the 25% dropout rate, we planned to register a total of 48 participants by calculating the number of subjects for each phase as 24. See also the reviewer’s comment #4 for an effect size.

Page 5, Line 174: The sample size was estimated to be 48 subjects yielding 80% power to detect an effect (Cohen's d=0.47) with a two-tailed t-test at a 0.05 significance level. The estimated change of the objective TST was based on the previously published data [42].

Reference >

42.        Howatson, G.; Bell, P.G.; Tallent, J.; Middleton, B.; McHugh, M.P.; Ellis, J. Effect of tart cherry juice (Prunus cerasus) on melatonin levels and enhanced sleep quality. European journal of nutrition 2012, 51, 909-916.

2. I generally encourage the authors to apply multiple testing corrections, e.g. by calculating the corresponding FDR value for each primary outcome test. This can easily be done in R based on the derived p values.

Author response >

As the reviewer recommended, we conducted additional multiple comparison correction analysis and described the details in the Methods. There were some variables with statistical significance even after multiple comparison corrections using Storey’s false discovery rate (q-value < 0.05 or < 0.01), which were marked as *, ** in the tables and figures, and the specific q-values were described in the Results section of the main text as follows.

Page 5, Line 185: Corrected p-value (or q-values) was also calculated using Storey’s false discovery rate (FDR) approach (95% confidence intervals) to correct for multiple comparisons.

Page 7, Line 215: *q-value < 0.05 and **q-value < 0.01, pFDR (Storey’s positive false discovery rate) was calculated as to account for multiple testing

Page 8, Line 243: *q-value < 0.05 and **q-value < 0.01, pFDR (Storey’s positive false discovery rate) was calculated as to account for multiple testing.

Page 9, Line 248: *q-value < 0.05 and **q-value < 0.01, pFDR (Storey’s positive false discovery rate) was calculated as to account for multiple testing.

Page 9, Line 258: *q-value < 0.05 and **q-value < 0.01, pFDR (Storey’s positive false discovery rate) was calculated for group*time difference by Linear mixed-effect model as to account for multiple testing.

3. Table 1 shows demographic characteristics of the study sample. For descriptive statistics I recommend to provide standard deviations or ranges instead of standard errors.

Author response >

We have modified to the standard deviations (SD) in the result section of main text and Table 1.

Page 5, Line 178: At baseline, participant characteristics were described as mean and standard deviation (SD) for continuous variables and frequency for categorical variables.

Page 6, Line 198: Data are mean ± SD (standard deviation) or answered numbers (%).

4. The authors should provide effect size estimates (e.g. eta squared), at least for the significant primary outcome results.

Author response >

Effect size estimates were performed as we described earlier under the reviewer’s first comment on a sample size. Please see the above and also the revised manuscript on page 5, line 174-176 with the reference #42.

5. Figure 2 appears to show a significant impairment of subjective sleep efficiency in the placebo phase. However, the authors state that ‘placebo phase failed to show any effects’.    The authors should reconsider the effect of time in the placebo phase and may provide an interpretation why sleep efficiency possibly got worse (‘by chance’ could be one explanation).

Author response >

Sorry for making this kind of confusion. Part of the reason for such confusion was from the previous sentence to describe the differences based on the analysis of the comparison between the test and placebo phases. To avoid this kind of misunderstanding, we have made the following corrections for the description in the revised manuscript as follows.

And we modified and supplemented the discussion section about the discrepancy between subjective and objective measures of sleep in terms of the behavioral intervention effect of the clinical trial including sleep diary.

Page 9, Line 252: Sleep diary showed a constant improvement in the sleep efficiency for the ACH phase group, compared with a gradual deterioration for the placebo group.

Page 10, Line 315: Our data regarding subjective sleep complaints measure – PSQI, ISI, and other subjective symptom scales - also exhibited gradual improvements during the ACH phase but did not show a statistically significant difference compared to the placebo phase. Subjective symptom scales in a double-blind trial could be insufficient to exhibit a significant difference between the placebo and tested ingredient due to placebo response. Also, it is possible that the behavior of keeping a sleep diary itself may have affected the subjective sleep and psychiatric parameters because it is the first stage of the clinical intervention of chronic insomnia [50].

Reference >

50.        Lund, H.G.; Rybarczyk, B.D.; Perrin, P.B.; Leszczyszyn, D.; Stepanski, E. The discrepancy between subjective and objective measures of sleep in older adults receiving CBT for comorbid insomnia. Journal of clinical psychology 2013, 69, 1108-1120.

6. Table 3, minor issue: A more appropriate caption might be ‘comparison of sleep parameters […]’ instead of ‘differences in sleep parameters […]’

Author response >

We have modified the description as below by referring to the comments.

Page 7, Line 240: Table 3. Comparison of sleep parameters between control (Placebo) and test (Alpha-s1 casein hydrolysate (ACH)) groups.

7. Table 4: The authors may define ‘improved sleep’ more clearly. Was there a certain degree of improvement necessary or did the direction of change – irrespective of magnitude – constitute the decisive criterion?

Author response >

The ‘improved sleep’ referred to the direction of the change and did not mean that the specific criterion should be met beyond a given number. The variables given in the table are described in detail as follows.

Page 7, Line 226: The percent ratio of participants with the direction of improved sleep quantity was also higher in the ACH phase compared to the placebo phase in both sleep diary and actigraphy measures.

Page 8, Line 248: Increased TST, Decreased SL, Increased SE, Decreased WASO, Decreased Awakes

8. Although the manuscript is generally well written, I feel that the grammar and style of the discussion section could be further improved. Some examples for odd sentences:

1) This study is the first clinical, randomized, placebo-controlled trial with both double-blind and crossover design’ – the first addressing sleep effects of ACH?

2) There were some simple double-blind studies revealed the efficacy ACH ingredient by PSQI improvements […].’

3) Because it was a method of measuring subjective symptoms in a double-blind trial, the difference between the two groups may not have been significant due to placebo response.’

4) There was one clinical ACH trial with both sleep diary, and actigraphy measurements showed beneficial effects concerning TST and efficiency; […]’

Regarding the discussion section, the authors may consider another English native speaker for proof-reading.

Author response >

We have modified the description as below by referring to the comments. And we are willing to have our manuscript apply for English editing service once the major revision of our paper has been approved.

Page 15, Line 306: The strengths of this randomized placebo-controlled trial are both double-blind, cross-over design and thorough sleep investigation by subjective and objective methods; moreover, it was conducted with a refined ACH capsule with relatively high dosage (300mg/day). So far, some studies were verifying the efficacy of the ACH without cross-over design only by subjective symptom scales [32,33]. There was one clinical ACH trial with both sleep diary and actigraphy measurements showed beneficial effects concerning TST and efficiency; however, their tested component was casein hydrolysate-enriched milk combined with other ingredients [34]. This is not only important to determine whether the specific ACH ingredient is beneficial for sleep; it can also be an important result for many adults with lactose intolerance who cannot drink milk [49].

Our data measured by subjective symptom scales - PSQI, ISI, daytime functioning and mood questionnaires - also exhibited gradual improvement during the ACH phase but did not show a statistically significant difference compared to the placebo phase. Subjective symptom scales in a double-blind trial could be insufficient to exhibit a significant difference between the placebo and tested ingredient due to placebo response. Also, it is possible that the behavior of keeping a sleep diary itself may have affected the subjective sleep and psychiatric parameters because it is the first stage of the clinical intervention of chronic insomnia [50].

Round  2

Reviewer 1 Report

The authors have adequately addressed most of my concerns raised for the previous version. Some minor suggestions:

1. Line 53-55, perspectives regarding how tryptophan and B vitamins may improve sleep among humans should be further discussed. It is not enough to solely mention their roles as precursors of serotonin, which only explains the possible mechanisms in an indirect way.

2. Line 309-313, please replace ‘subjective symptom scales’ with the specific questionnaires used in this study (i.e. PSQI, ISI, etc.).

Author Response

< Reviewer 1 >

1. Line 53-55, perspectives regarding how tryptophan and B vitamins may improve sleep among humans should be further discussed. It is not enough to solely mention their roles as precursors of serotonin, which only explains the possible mechanisms in an indirect way.

Author response >

Thank you for pointing out an important part. We added the sentence regarding the direct effect of tryptophan on the sleep regulation.

Page 2, Line 53: Tryptophan has been shown to have direct beneficial effects on the homeostatic regulation of sleep, not only in subjects with insomnia but also those without sleep disturbance have been found to increase sleepiness and shorten sleep latency [14]. Moreover, unlike the photoperiodically regulated production of melatonin in the pineal gland, the release of gastrointestinal melatonin by the enterochromaffin cells appears to be regulated by food intake, particularly after tryptophan loading. [15].

Reference>

14.        Silber, B.; Schmitt, J. Effects of tryptophan loading on human cognition, mood, and sleep. Neuroscience & Biobehavioral Reviews 2010, 34, 387-407.

15.        Bubenik, G.A. Gastrointestinal melatonin: localization, function, and clinical relevance. Digestive diseases and sciences 2002, 47, 2336-2348.

2. Line 309-313, please replace ‘subjective symptom scales’ with the specific questionnaires used in this study (i.e. PSQI, ISI, etc.).

Author response >

Thank you for advising to correct the sentence in a clear way. We have modified the description as below by referring to the comments.

Page 10, Line 312: Our data, measured by subjective symptom scales including PSQI, ISI, ESS, FSS, BDI, and BAI, also demonstrated gradual improvement during the ACH phase but did not show a statistically significant difference compared to the placebo phase.

Reviewer 2 Report

The authors adequately addressed all my concerns. One minor issue left: I feel that it should not be overgeneralized that, in a double-blind trial, subjective symptom scales may be insufficient to show significant differences between placebo and tested ingredient. If there is a true effect of tested ingredient on subjective symptoms that is greater than placebo, than it is just a question of sample size to render this effect visible. If this true effect is not greater than placebo, then the tested ingredient obviously has no effect on subjective symptoms. The proposed sentence might be modified:

Line 309, original:

Subjective symptom scales in a double-blind trial could be insufficient to show a significant difference between the placebo and the tested ingredient due to the placebo response.

Suggested modification:

"Hence, in the present double-blind trial with 48 participants, potential effects of ACH on applied subjective symptom scales appeared to be too marginal to exceed the placebo response.

I do not have any further comments.

Author Response

< Reviewer 2 >

1. The authors adequately addressed all my concerns. One minor issue left: I feel that it should not be overgeneralized that, in a double-blind trial, subjective symptom scales may be insufficient to show significant differences between placebo and tested ingredient. If there is a true effect of tested ingredient on subjective symptoms that is greater than placebo, than it is just a question of sample size to render this effect visible. If this true effect is not greater than placebo, then the tested ingredient obviously has no effect on subjective symptoms. The proposed sentence might be modified:

Line 309, original: Subjective symptom scales in a double-blind trial could be insufficient to show a significant difference between the placebo and the tested ingredient due to the placebo response.

Suggested modification: "Hence,in the present double-blind trial with 48 participants, potential effects of ACH on applied subjective symptom scales appeared to be too marginal to exceed the placebo response.

Author response >

Thank you for pointing out the important part of that we should not over-generalize our results. We have modified the description as below by referring to the comments.

Page 10, Line 315: Potential effects of ACH on applied subjective symptom scales might be too marginal to exceed the placebo response in the present double-blind trial with 48 participants.
